# Memory phase-specific genes in the Mushroom Bodies identified using CrebB-target DamID

Noemi Sgammeglia[1], Yves F. Widmer[1], Jenifer C. Kaldun[1], Cornelia Fritsch[1], Rémy Bruggmann[2], Simon G. Sprecher[1]*

1 Department of Biology, University of Fribourg, Fribourg, Switzerland, 2 Interfaculty Bioinformatics Unit and Swiss Institute of Bioinformatics, University of Bern, Bern, Switzerland

* simon.sprecher@unifr.ch

## Abstract

The formation of long-term memories requires changes in the transcriptional program and *de novo* protein synthesis. One of the critical regulators for long-term memory (LTM) formation and maintenance is the transcription factor CREB. Genetic studies have dissected the requirement of CREB activity within memory circuits, however less is known about the genetic mechanisms acting downstream of CREB and how they may contribute defining LTM phases. To better understand the downstream mechanisms, we here used a targeted DamID approach (TaDa). We generated a CREB-Dam fusion protein using the fruit fly *Drosophila melanogaster* as model. Expressing CREB-Dam in the mushroom bodies (MBs), a brain center implicated in olfactory memory formation, we identified genes that are differentially expressed between paired and unpaired appetitive training paradigm. Of those genes we selected candidates for an RNAi screen in which we identified genes causing increased or decreased LTM.

## Author summary

CREB dependent regulation of gene expression is a key step in long-term memory formation and consolidation. Using a Dam::CREB fusion protein we identified genes that are differentially expressed in the mushroom body one day and two days after appetitive olfactory conditioning. We found candidate genes that cause memory enhancement and genes that cause memory suppression. Thus, we identified potential new regulators of memory formation and maintenance.

## Introduction

The capability to form memories is an important feature of the brain that allows the adaptation to a dynamic environment based on past experiences. In contrast to short-term memory (STM), enduring forms of memory, referred to as long-term memory (LTM), require the activation of specific transcriptional programs which, ultimately, leads to "de novo" protein

**Data Availability Statement:** All data is included in the manuscript. Sequencing data are deposited and accessible (GEO accession number GSE216045).

**Funding:** The current work is supported by the Swiss National Science Foundation grant 310030_188471 and grant CRSII5_180316 to SGS. The funders had no role in study design, data collection and analysis, decision to publish, or preparation of the manuscript.

**Competing interests:** The authors have declared that no competing interests exist.

synthesis [1–4]. This, in turn, enables structural and functional rearrangements at the level of synapses that are critical for plasticity [5–7]. Understanding the molecular mechanisms behind synaptic plasticity and the establishment of memory phases remains one of the major objects of study in neuroscience. The cAMP response element-binding protein (CREB) is a well conserved transcription factor (TF) which regulates different biological processes, including development and plasticity [8–11]. Studies on learning and memory have revealed that induction of CREB acts as important activator for LTM formation [9,12,13]. Following LTM-training, CREB is activated by phosphorylation, downstream of the cAMP/PKA cascade [14,15]. Learning-induced neuronal excitability triggers this signaling pathway and, conversely, interfering with elements of this pathway selectively impairs the expression of LTM [16,17]. CREB also requires the interaction with certain co-activators and epigenetic factors such as histone acetyl transferases (HATs) and DNA methylases [18,19], to regulate the transcription of target genes in specific phases of memory formation and maintenance.

In the fruit fly *Drosophila melanogaster*, two CREB proteins are encoded by *CrebA* and *CrebB* genes [20,21]. While *CrebA* only recently has been linked to LTM formation [22], many experimental evidences have shown the involvement of *CrebB* within learning and memory circuits, and confirmed its pivotal role in gating protein synthesis-dependent long-term memories [12,16,17,19,23–27].

Several genetic screens in *Drosophila* have contributed to the identification of genes involved in specific phases of memory establishment [28–30]. Many of these genes enable memory formation ("memory promoter genes"), whereas others negatively interfere with it ("memory suppressor genes") [30]. Whole-head transcriptional studies have provided interesting insights about LTM transcriptional changes [31,32] and significant advantage has been yielded restricting the analysis to the Mushroom Bodies (MBs) [29,33–36].

The MBs constitute a main centre for olfactory associative memory formation [37]. The olfactory information is acquired from the olfactory receptors, located in the antennae, and passed to the glomeruli where projecting neurons forward it to the Kenyon Cells (KCs). There are ~2500 KCs per hemisphere which project their axons to the mid brain, forming a pair of L-shaped neuropils [38]. The α β and α' β' KCs projections form the vertical and horizontal lobes, whereas the γ KCs projections only form the horizontal lobes of MBs [39]. Downstream to the KCs, MB output neurons (MBONs) integrate the olfactory information and deliver it outside the MBs [38]. Dopaminergic neurons from specific clusters provide the negative or positive reinforcement during conditioning training interfering at the level of KC-MBON synapses [40]. The updated information is then forwarded to the motor centers for the execution of a proper learning-induced behavioral response. As a regulator of LTM formation, CREB activity is crucial within the MB circuitry. However, its requirement among different subsets of MB neurons tends to vary according to the type of memory. During appetitive LTM memory formation, for instance, CREB is required in α/β and α'/β' lobes and it seems dispensable in γ lobes [17]. Water-reward LTM formation requires CREB only in α/β surface and γ dorsal neurons [27]. A possible explanation of the different implication of CrebB is that its activity relies also on other co-factors, which may be differentially distributed within the MB circuits.

While many studies have shown the importance of CREB in different MB sub-compartments during LTM consolidation, less is known about CREB targets and their actual impact on the shift of LTM phases. A previous study used the ChIP-seq technique to investigate genes differentially regulated in MB nuclei during LTM early maintenance [19]. To specifically address this phase, they decided to map CRTC binding in proximity of CREB target sites, following 1-day spaced training. Contrary to CBP, CRTC is dispensable for LTM formation but required during early maintenance [19,25]. Candidate genes for LTM maintenance were extracted considering the overlap of CRTC/CREB binding with two histone acetyltransferases

(HATs), GCN5 and Tip60, acetylation. These two HATs, seem to be dispensable for LTM formation but necessary for 4-day LTM maintenance, promoting the expression of *Beadex* and *Smrter*. In a later stage CRTC/CREB and GCN5 are no longer required. Tip60 and Beadex, instead, are necessary for 7-day LTM maintenance.

Studying LTM-induced differentially regulated genes constitutes an entry point to understand the molecular mechanisms that regulate each phase of memory consolidation and its maintenance. To investigate the CrebB-directed transcriptional changes during LTM we employed, in this study, the targeted DamID (TaDa) technique [41–43]. TaDa enables gene expression profiling with spatial and temporal control. This procedure is based on the employment of a DNA adenine methyltransferase (Dam) from *Escherichia coli* fused to an RNA polymerase or a transcription factor. The expression of the fusion protein is followed by the methylation of adenine in the sequence GATC in nearby loci, providing a readout of the transcriptional program. By sequencing the methylated fragments, it is possible to access the target genes. Here we created a CrebB-Dam fusion protein to identify CrebB target genes within two different memory time intervals. Following data analysis, we extracted lists of CrebB candidate target genes during 2 time intervals (TI-1 and TI-2) and performed knockdown experiments for these candidates, testing 24h and 48h memory performance, respectively. From both TIs we identified potential "memory suppressor" and "memory enhancer" CrebB targets, which we further characterized for their memory phase specificity comprehensively at 0h, 24h and 48h. Unc-5 was selected for additional memory tests within the lobes of the MBs, showing its involvement in all of the LTM phases.

## Results

### CREB dependent gene expression profiles during LTM formation intervals

To build a link between transcriptomic and functional LTM changes we set out to profile the temporal expression of CrebB-target genes in the Mushroom Bodies, following associative training. We used a modified version of the DNA adenine methyltransferase identification (DamID) technique, referred to as Targeted DamID (TaDa) sequencing [41,42]. The principle behind DamID is to profile genome-wide DNA binding of a DNA-associated protein of interest by fusing it to an *E. coli* DNA adenine methyltransferase (Dam), which methylates adenine in surrounding GATC sequences. TaDa technique takes advantage of the Gal4/UAS binary system and its repressor (Gal80ts) to control, temporally, the expression of the fused proteins in a cell- or tissue-specific fashion [42,44]. In this study, we generated a Dam::CrebB fusion protein (UAS-Dam::CrebB) to identify CrebB targets during LTM formation. The synthetized Dam::CrebB protein maintains the ability to bind specifically to the CRE sequences, as confirmed by EMSA analysis (S1 Fig). Using the mb247-Gal4 driver, we induced Dam::CrebB expression in the α, β, and γ lobes of the MB [45] (Fig 1A). Due to the dicistronic nature of the Dam::CrebB transgene [42], the fusion protein is expressed at very low levels. In addition, to ensure that the DAM::CrebB fusion protein does not interfere with normal memory formation we performed classical olfactory conditioning with those lines and did not observe any impairment (S2 Fig).

TaDa was performed on flies conditioned to form long-term memory in a classical olfactory paradigm, where sucrose is used as positive reinforcer. During the training phase, starved flies were sequentially exposed to two odors, one of which was paired with sucrose (paired training). A single cycle of appetitive learning is able to induce the formation of LTM [46,47]. As control, we included the unpaired group, where odor and sucrose were presented to the flies temporally separated, preventing odor-reward association to be installed. An additional control group (naïve) consisted of wild type flies which were raised and maintained under the

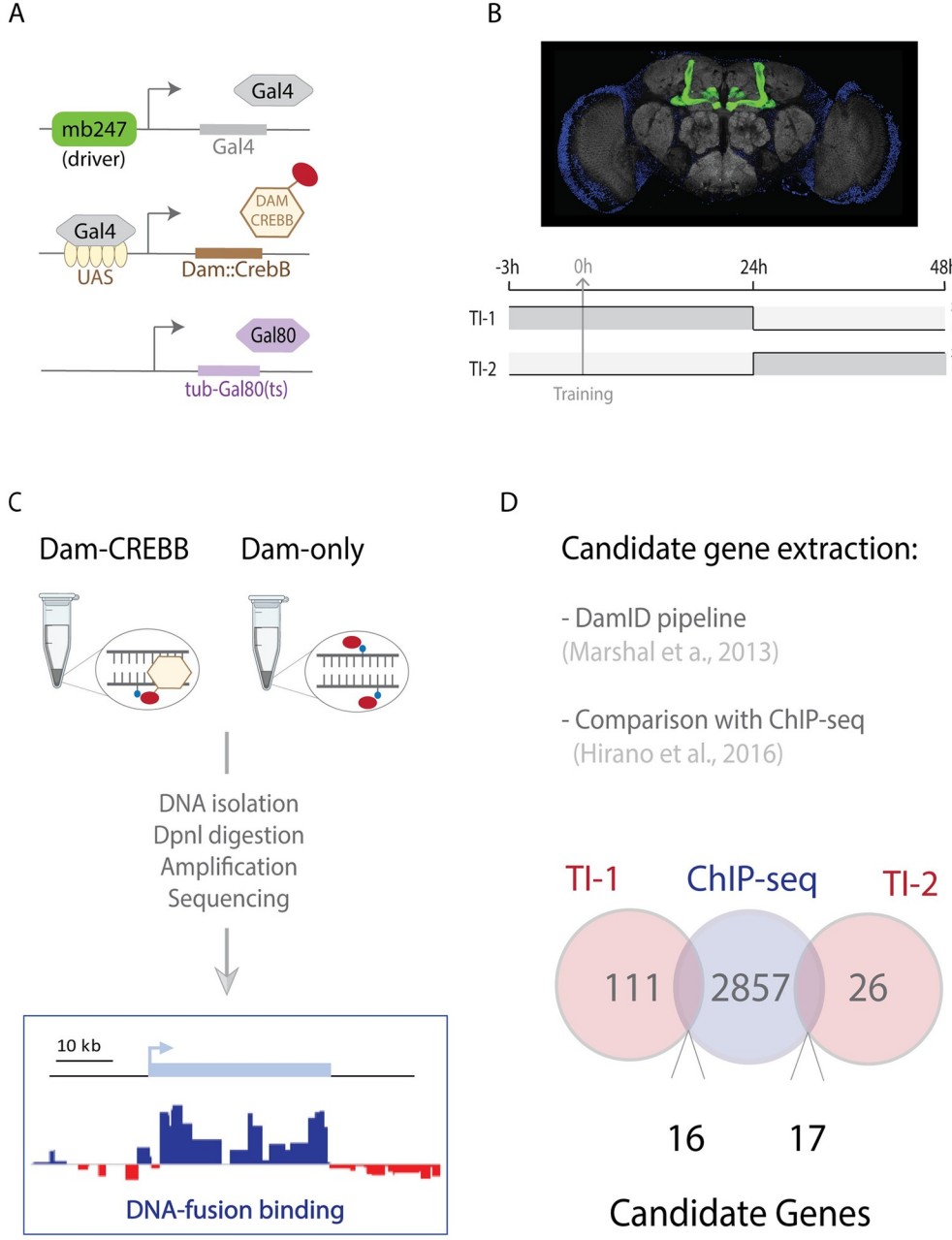

**Fig 1. CREBB-regulated transcriptomic changes during LTM phases.** (A) Illustration of the Gal4/UAS binary system lines used in the TaDa experiment. The driver mb247 restricts the expression of the transcription factor Gal4 to the Mushroom Body cells. Gal4 binds to the UAS enhancer and promotes the expression of the CrebB-Dam fusion protein. The temperature sensitive Gal80ts inhibits Gal4 at permissive temperature (18–25˚C) and it is inactivated at 29˚C. (B) Above, brain dissected from the mb247-reporter line (mb247-Gal4>UAS-myr::GFP, GFP in green). Below, schematic illustration of the experimental design where the temporal intervals are defined by the switch of the temperature from 18˚C to 29˚C, enabling the transcription of the fused protein. The grey arrow indicates the time of conditioning session (training). The two red arrows indicate the time point at which the heads were collected for DNA extraction. (C) Schematic representation of the Targeted DamID (TaDa) pipeline. Dam-CrebB and Dam-only expression was induced in the Mushroom Body. Genomic DNA was extracted from *Drosophila* heads and digested with the methylation sensitive restriction enzyme Dpnl. Methylated fragments were PCR amplified and sequenced. The extracted reads were mapped to a reference genome and the log2 ratio of Dam-Creb/Dam-only was calculated. (D) The candidate genes of TI-1 and TI-2 (light red circles) were extracted through the DamID pipeline and compared to CREB-targets (light blue circle) obtained from a previous ChIP-seq extraction [19].

same condition (food and temperature) but were not exposed to the paired nor the unpaired training.

To temporally dissect the transcriptional changes that lead to LTM formation and maintenance, we performed CREB-target DamID at two different time intervals (TIs) following LTM formation in the MBs. The temporal restriction was achieved by shifting the flies to 29°C, thus, inactivating the temperature sensitive tubulin-Gal80$^{ts}$ (Gal4 inhibitor) and allowing the induction of Dam::CrebB or Dam-only expression [44]. This system was applied in two different time intervals (TIs). Time interval 1 (TI-1) spanned from 3h before the associative training (0h) to 24h after. Time interval 2 (TI-2) started from 24h after the associative training and ended at 48h (Fig 1B). While TI-1 embrace memory formation and consolidation, TI-2 matches with its early maintenance. Three biological replicates were used for experiment and control (Dam::CrebB and Dam-only) as well as for each of the three conditions (paired, unpaired and naïve), for each of the two TIs. After the period of induction, the heads of the flies were collected. Genomic DNA was extracted and digested with DpnI, a restriction enzyme which cuts at adenine-methylated GATC sites. Methylated fragments were PCR amplified and sequenced using Illumina HiSeq3000 (Fig 1C).

To assess the overall binding profile, we processed sequencing data with the DamID-seq pipeline [48]. We compared Dam::CrebB with Dam-only methylation profile, assigning a weighted log2 ratio where positive values indicate that the GATC sites are preferentially methylated by Dam::CrebB compared to background methylation (Dam-only). Next, we compared the methylated reads of the paired condition with the unpaired, to extract a list of genes specifically correlated to the learning paradigm. Significant genes were assessed using the log2 fold-change with a cut off of 0.2 and a False Discovery Rate (FDR)-adjusted p-value (q-value) of 0.01. To conduct a more stringent selection we excluded from the list of the learning-induced (paired) genes the ones that where enriched in the naïve control group, obtaining 111 genes for TI-1 and 26 genes in TI-2 (S1 File). Finally, we decided to compare our candidate genes with a ChIP-seq dataset published previously [19], which has provided the binding coordinates of genes targeted by CrebB and its cofactor CRTC, following aversive learning. 33 of the genes found in our study were also present in the above-mentioned ChIP-seq data and for this reason selected for further experiments. Specifically, 16 of these genes are differentially regulated in TI-1 and 17 in TI-2 (Fig 1D).

## Candidate RNAi screen identifies *HERC2, cic, unc-5 and esn* as CrebB targets

To test the potential involvement of the 33 candidate genes in learning and memory functions we performed a knockdown screen and assessed the memory performance of corresponding RNAi lines. For each of the candidates, UAS-RNAi constructs were expressed under the control of the mb247-Gal4 driver. The list of candidates obtained in the TaDa analysis was further restricted during the RNAi screen, considering the availability of RNAi lines, the viability and the number of the MB-Gal4>UAS-RNAi offspring. TI-1 genes included in the screen were: *Adenylyl cyclase 13E* (*Ac13E*), *BIR repeat containing ubiquitin-conjugating enzyme* (*Bruce*), *C3G guanyl-nucleotide exchange factor* (*C3G*), *CG11873*, *CG43134*, *capicua* (*cic*), *HECT and RLD domain containing protein 2* (*HERC2*), *Nuclear receptor coactivator 6* (*Ncoa6*), *no circadian temperature entrainment* (*nocte*), *sloppy paired 1* (*slp1*), *Serendipity δ* (*Sry-δ*) and *visceral mesodermal armadillo-repeats* (*vimar*). TI-2 selected genes were: *6-phosphofructo-2-kinase* (*Pfrx*), *espinas* (*esn*), *bves*, *CG30419*, *Moesin* (*Moe*), *mrj*, *TAK1-associated binding protein 2* (*Tab2*), *CG10444*, *karmoisin* (*kar*), *Coronin* (*coro*), *unc-5*. Flies expressing the RNAi of TI-1 genes were tested for 24h LTM and flies expressing the RNAi of TI-2 genes were tested for 48h

LTM. The UAS-*Dcr2* transgene was co-expressed to increase the efficiency of the knockdown machinery [49]. Using the same appetitive paradigm, as described above, previously starved flies were trained to associate an odor with rewarding sucrose. Since the different RNAi lines used in this assay had been generated in different genomic backgrounds we used fly-lines with a similar genomic background for the respective RNAi lines as controls. For this reason, we grouped the RNAi-lines in $w^{1118}$-, $y^1w^1$-, $y^1v^1$- or $y^1sc*v^1sev^{21}$- background lines according to their genomic background and compared their memory performance with flies of the same genotype but without UAS-RNAi-transgene. In addition, we included the UAS-RNAi/+ control for each UAS-RNAi line. For the assay, the RNAi and control lines were crossed to the driver line (UAS-*Dcr2*;;mb247-Gal4). Of the 23 RNAi lines tested, 4 genes showed significant LTM phenotypes (Fig 2). Specifically, among the lines tested at 24h, *HERC2*-RNAi (Fig 2A) showed an increased learning index (LI) score (P-value 0.002) whereas *cic*-RNAi (Fig 2E) displayed a decreased performance (P-value 0.04). Of the lines tested at 48h, *esn*-RNAi (Fig 2B) showed increased performance (P-value 0.02) and *unc-5*-RNAi (Fig 2G) a decreased score compared to the respective control (P-value 0.03). Our experiments have identified candidate genes which are potential regulators of learning and memory mechanisms, either as "memory enhancer" or "memory suppressor" genes [30]. To validate the knockdown efficiency, we have performed a qPCR analysis on whole fly heads, pan-neuronally knocked down for each gene (Elav-Gal4 x UAS-RNAi), and compared to controls (Elav-Gal4/+ and +/UAS-RNAi). Of the 22 genes tested 3 were lethal with the elav-Gal4 driver and 13 showed reduced mRNA expression compared to their controls. The remaining 6 genes did not show a reduction compared to both controls, probably due to inefficient knockdown, or a higher expression in those cells that were not addressed by our pan-neuronal Gal-4 line (e.g. glial cells). In many cases the +/UAS-RNAi control already shows a reduction of RNA levels compared to the Elav-Gal4 control suggesting that there is some leaky expression of these UAS-RNAi constructs (S3 Fig).

### Assessing memory phase specificity of the candidate genes

To dissect the potential involvement of the four candidate genes into specific memory phases, we tested them at different time points (Fig 3). In particular, *HERC2* and *cic* (TI-1 genes) where further tested at 0h and 48h and *esn* and *unc-5* (TI-2 genes) at 0h and 24h. The *HERC-2*-RNAi phenotype seems to be exclusive for 24h LTM. However, 48h LTM showed a visible trend of enhanced memory compared to control with a P-value only marginally higher than 0.05 (P-value 0.09) (Fig 3A). Similar to 24h LTM, *cic*-RNAi showed a decreased memory performance at 0h (P-value 0.01), but the 48h LTM score was not significant (Fig 3B). *Unc-5* knockdown showed a decreased LI score in all the three time points (0h P-value 0.007; 24h P-value 0.03; 48h P-value 0.02) (Fig 3C). Finally, the RNAi of *esn* showed an increased LI score for both 24h (P-value 0.0001) and 48h (P-value 0.02) LTM, but no significance at 0h memory (Fig 3D).

### HCR validation of gene expression in the MBs

We performed High Chain Reaction (HCR) *in situ* hybridization to assess the expression of the genes in MB reporter lines (UAS-myr::GFP/+; mb247-Gal4/+). Using designed probes, we visualized mRNA of genes and assessed their expression by detecting the fluorescent hairpins attached to their probes (Fig 3A', 3B', 3C' and 3D' and S4 Fig). While HERC2, cic, and unc-5 mRNA molecules were detected in MB cells (Fig 3A'-A", 3B'-B" and 3C'-C" and S4A-A", S4B-B" and S4D-D" Fig), in HCR analysis for esn mRNA only a small number of MB cells were stained positive for esn mRNA (Fig 3D'-D" and S4C-C" Fig). One possible explanation may be that esn expression in specific MB cells may only be transiently induced during LTM

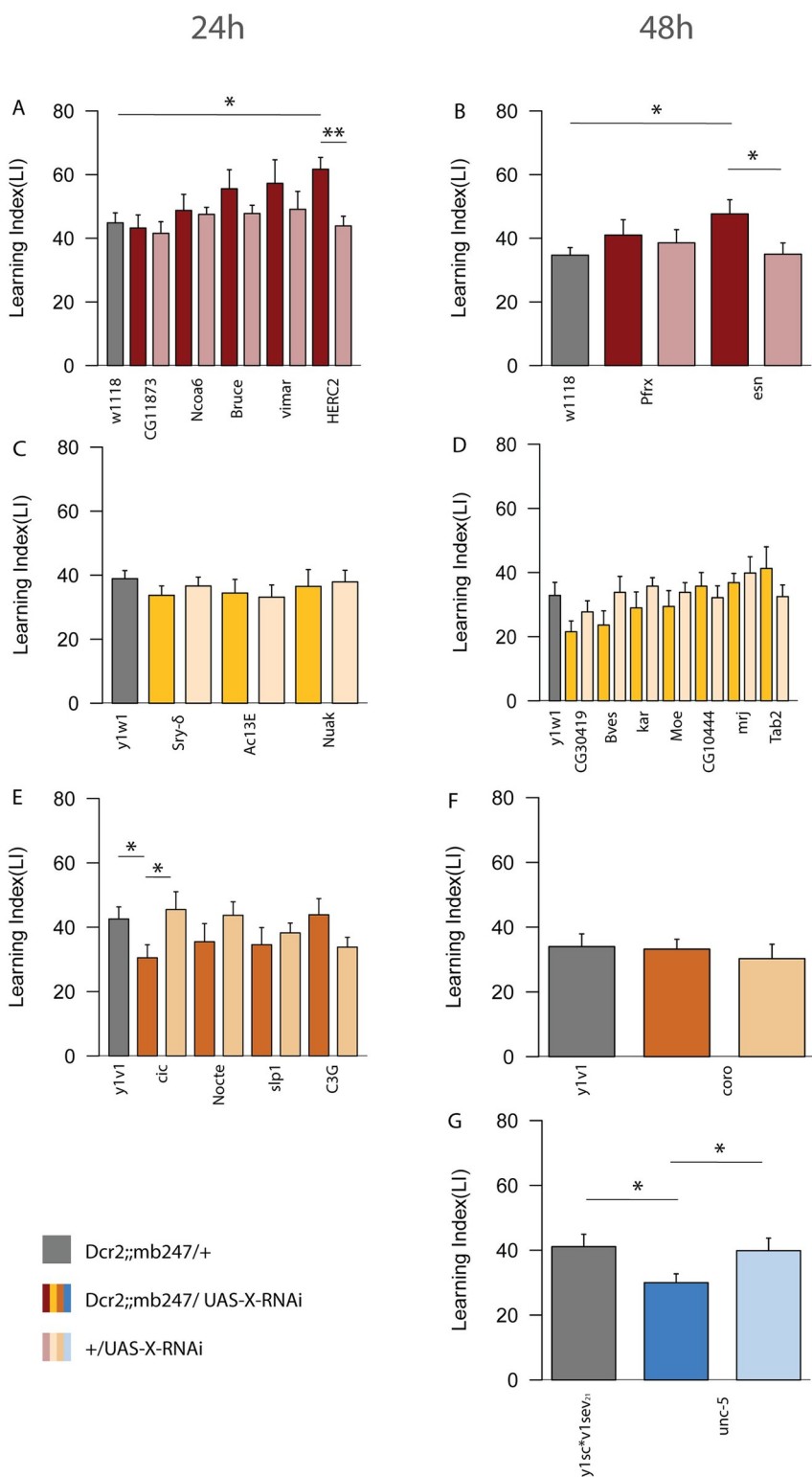

**Fig 2. 24h and 48h memory screen.** RNAi constructs were used to induce the transcriptional knockdown of the candidate genes from TI-1 and TI-2. Transgenic crosses carrying the RNAi system were grouped according to their genetic background ($w^{1118}/y^1w^1/y^1v^1/y^1sc^*v^1sev^{21}$) and tested for 24h (A, C, E) or 48h (B, D, F, G), following appetitive olfactory conditioning. Asterisks indicate the P-value of the comparison with control crosses (mb247-Gal4>$w^{1118}/y^1w^1/y^1v^1/y^1sc^*v^1sev^{21}$) (* P < 0.05, ** P < 0.01, *** P < 0.001); n = 7–10 for MB > RNAi, n = 7–28 for control crosses. Bar graphs represent the mean and error bars represent the SEM.

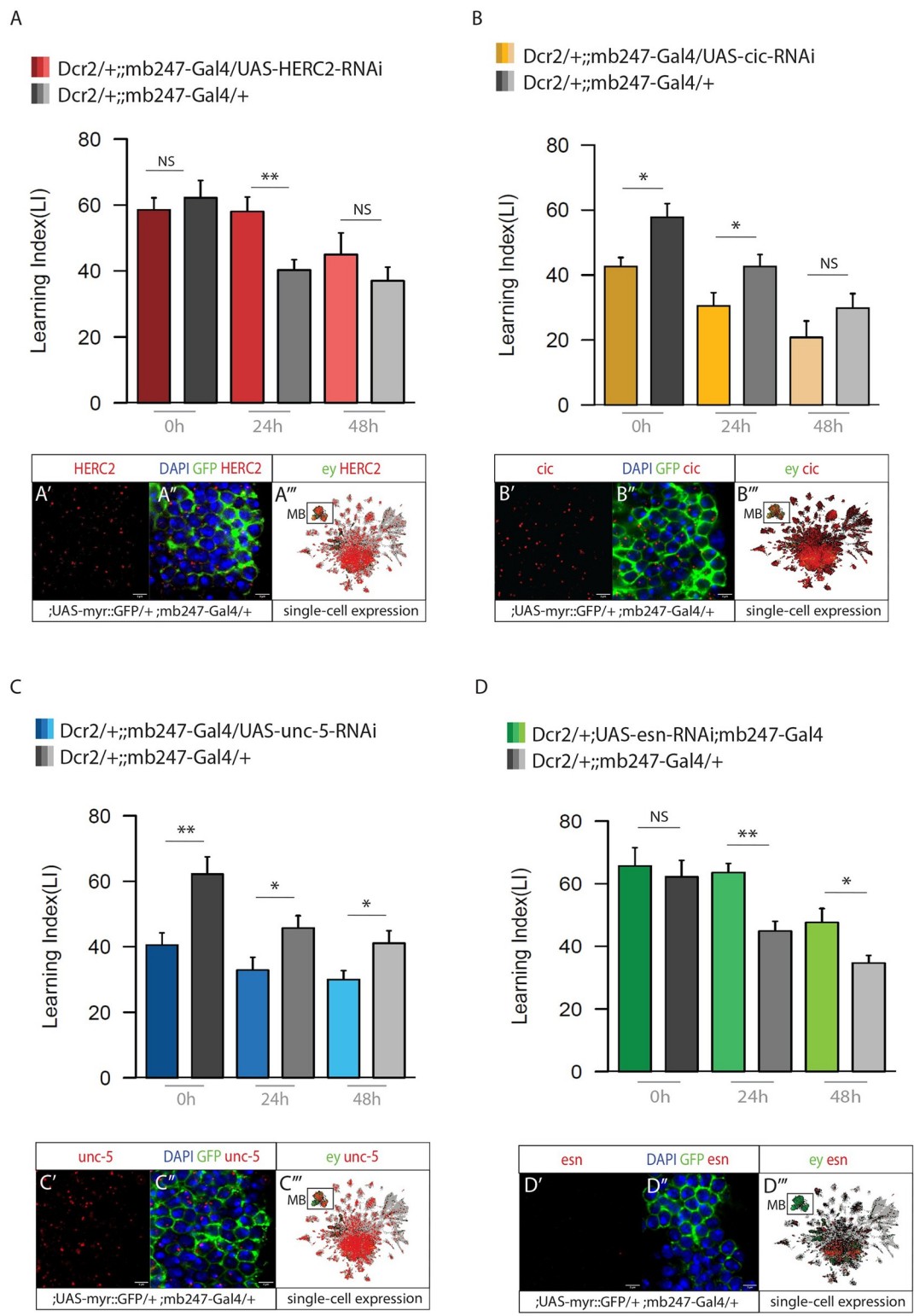

**Fig 3. Memory phases specificity.** CREB target genes with a significant memory phenotype were tested further at different time points, comparing 0h, 24h and 48h memory. (A) Memory performance for HERC2-RNAi in the MB, showing increased memory at 24h but not at 0h an 48h. (A', A") HCR expression of HERC2 mRNA in MB reporter line (;;mb247-Gal4 >; UAS-myr::GFP). (A''') HERC2 single-cell expression. (B) Memory performance for cic-RNAi in the MB, showing a significant decreased performance at 0h and 24h. (B', B") HCR expression for cic mRNA in MB reporter line. (B''') cic single-cell

expression. (C) Memory performance for unc-5-RNAi in the MB, showing a defect in all the 3 time points. (C', C") HCR expression of unc-5 mRNA in MB reporter line. (C"') unc-5 single-cell expression. (D) Memory performance for esn-RNAi in the MB, showing a decreased LTM both at 24h and 48h. No defect at 0h. (D', D") HCR expression (not detected) of esn mRNA in MB reporter line. (D"') esn single-cell expression. n = 7–10 for MB > RNAi and control crosses. Bar graphs represent the mean and error bars represent the SEM. Asterisks denote significant difference between groups (* P < 0.05, ** P < 0.01, *** P < 0.001).

formation. Our results are consistent with the expression profile reported on the single-cell transcriptome atlas of the *Drosophila* adult brain [50] (Fig 3A"', 3B"', 3C"' and 3D"'). Here, MB cells are highlighted in green through the expression pattern of the MB-marker eyeless (ey). Single-cell expression of the candidate genes are reported in red.

## Adult-specific knockdown of *HERC2*, *esn*, *cic* and *unc-5*

To exclude the possibility that the memory phenotype observed in our RNAi experiment was due to developmental defects, we restricted the knockdown of the four candidate genes (*HERC2*, *esn*, *cic and unc-5*) to the adult stage (Fig 4). To achieve temporal control on the RNAi, we co-expressed the temperature sensitive tubGal80ts in the MBs, which inhibits Gal4 at the permissive temperature of 18˚C, but not at 29˚C. Thus, while the experimental crosses were kept at 18˚C, the derived offspring was shifted to 29˚C for ~4 days, before training and kept at this temperature till testing for LTM. The memory performance of the adult-specific knockdown was consistent with our previous observation. In particular, *HERC2* adult-specific RNAi in the MBs (UAS-Dcr2;tubGal80ts;mb247-Gal4/UAS-HERC2-RNAi) displayed increased 24h memory (Fig 4A), compared to control 1 (UAS-Dcr2;tubGal80ts;mb247-Gal4/+, P-value P ≤ 0.01) and control 2 (+/UAS-HERC2-RNAi, P-value P ≤ 0.01). Similarly, *esn* adult-specific RNAi showed an increased 48h memory performance (Fig 4B) compared to control 1 (UAS-Dcr2;tubGal80ts;mb247-Gal4/+, P-value ≤ 0.001) and control 2 (+/UAS-esn-RNAi, P-value ≤ 0.001). The adult-specific RNAi of *cic* showed a decreased 24h memory score (Fig 4C) compared to control 1 (UAS-Dcr2;tubGal80ts;mb247-Gal4/+, P-value ≤ 0.001) and control 2 (+/UAS-cic-RNAi, P-value ≤ 0.001). Finally, *unc-5* adult-specific RNAi tested for 48h memory displayed disrupted memory (Fig 4D) compared to control 1 (UAS-Dcr2;tub-Gal80ts;mb247-Gal4/+, P-value ≤ 0.001) and control 2 (+/UAS-unc-5-RNAi, P-value ≤ 0.001).

## Knockdown of *unc-5* in the MB lobes decreases LTM performance

We next decided to focus on *unc-5*, which encodes a netrin receptor involved in motor axon guidance and cell migration [51–54]. *unc-5* has previously been shown to be important for its role in synaptic plasticity, learning and memory formation in mammals and its ligand Netrin-B has been shown in *Drosophila* to be important for courtship memory [54–57]. We therefore decided to further investigate the requirement of unc-5 within the MB lobe system in more detail. To dissect *unc-5* activity within memory-related circuits, we tested 24h LTM of the RNAi of unc-5 in specific lobes of the MBs. We used the driver VT030604-Gal4 to guide *unc-5* knockdown in the α'/β' lobes; the c739-Gal4 driver to address the α/β lobes and the 5-HTR1B-Gal4 for the γ lobe. In all the MB subsets, *unc-5* knockdown showed decreased memory performance compared to control (Fig 5A). To further validate *unc-5* function in LTM, we used two other available UAS-RNAi lines and tested their memory performance. Both RNAi lines showed a decreased 24h memory score when compared to their respective genomic background controls (y$^1$w$^1$ and w$^{1118}$) (Fig 5B).

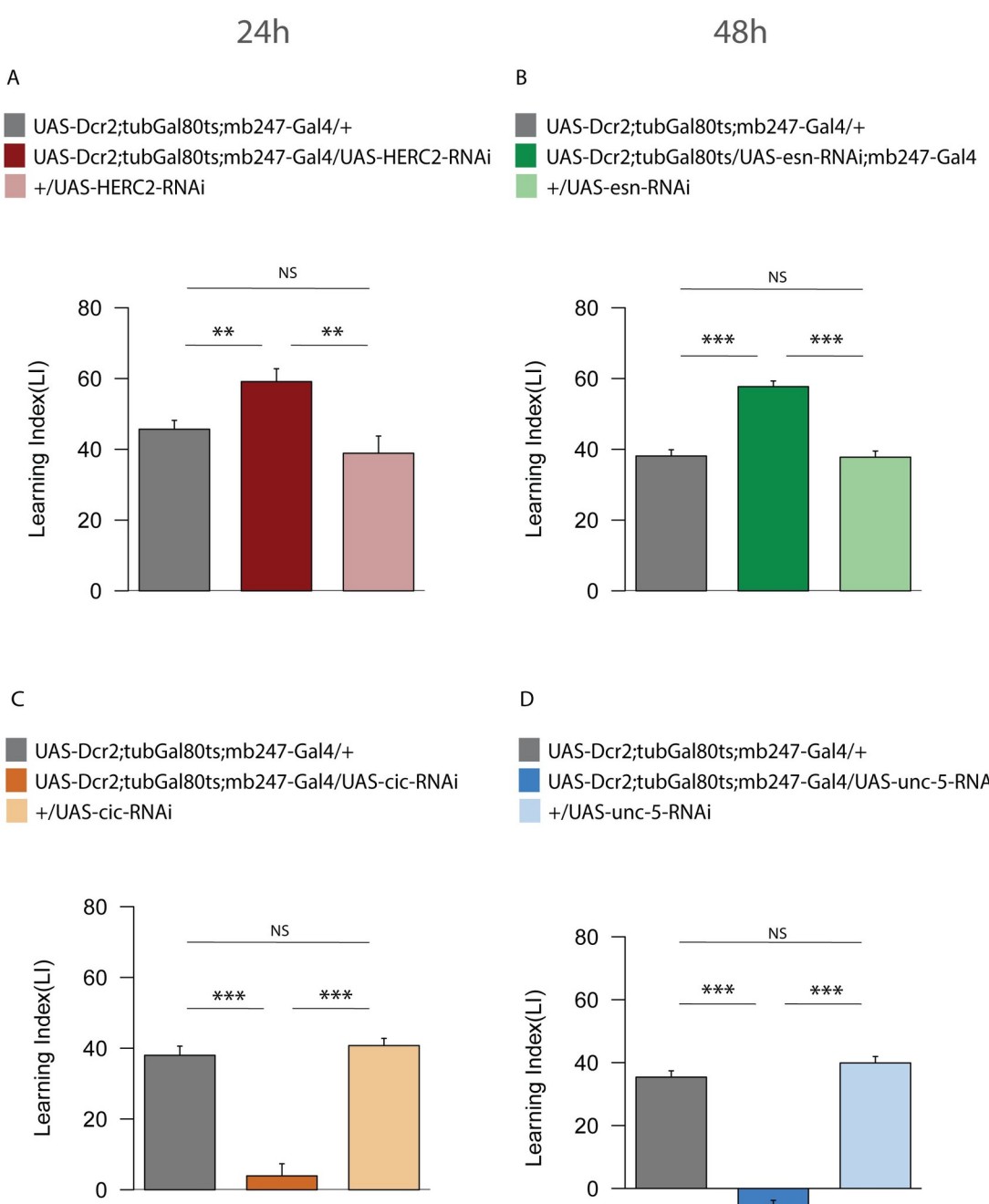

**Fig 4. Memory tests in adult-specific RNAi of HERC2, esn, cic and unc-5.** We co-expressed tubGal80ts in the MBs to temporally control the induction of HERC2, esn, cic and unc-5 RNAi. (A) 24h LTM performance of the adult-specific HERC2 RNAi, in the MBs. (B) 48h LTM performance of the adult-specific esn RNAi, in the MBs. (C) 24h LTM performance of the adult-specific cic RNAi, in the MBs. (D) 48h LTM performance of the adult-specific unc-5 RNAi, in the MBs. Bar graphs represent the mean and error bars represent the SEM. Asterisks denote significant difference between groups (* P < 0.05, ** P < 0.01, *** P < 0.001). Numbers signify P-values (Welch two sample t-test).

A

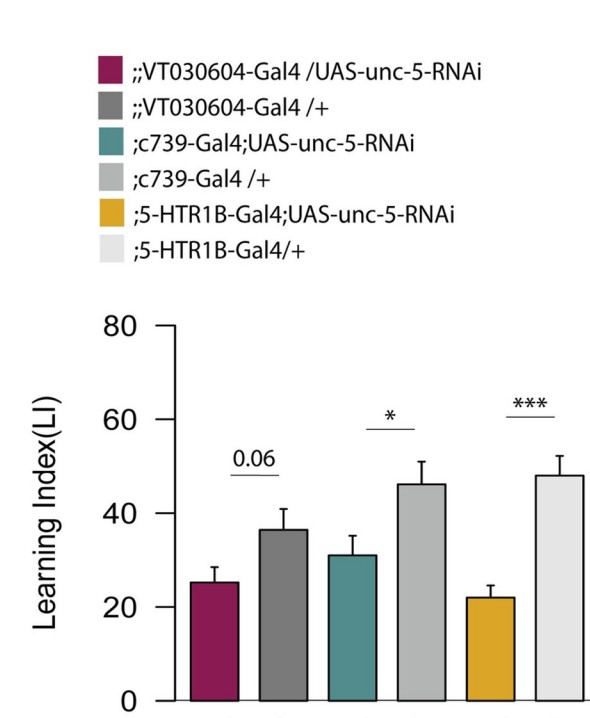

■ ;;VT030604-Gal4 /UAS-unc-5-RNAi
■ ;;VT030604-Gal4 /+
■ ;c739-Gal4;UAS-unc-5-RNAi
■ ;c739-Gal4 /+
■ ;5-HTR1B-Gal4;UAS-unc-5-RNAi
■ ;5-HTR1B-Gal4/+

B

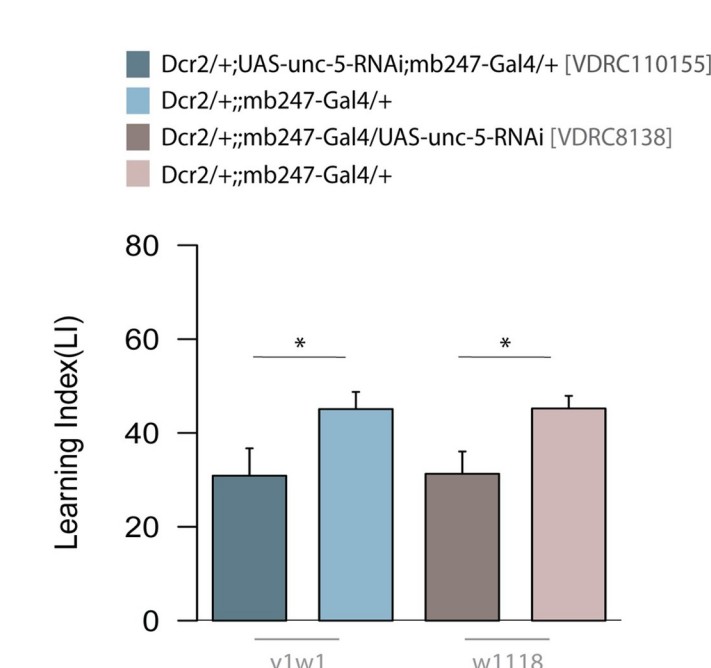

■ Dcr2/+;UAS-unc-5-RNAi;mb247-Gal4/+ [VDRC110155]
■ Dcr2/+;;mb247-Gal4/+
■ Dcr2/+;;mb247-Gal4/UAS-unc-5-RNAi [VDRC8138]
■ Dcr2/+;;mb247-Gal4/+

**Fig 5. *unc-5* knockdown in MB lobes and validation in two different RNAi lines.** (A) Using VT030604, c739 and 5-HTR1B drivers, unc-5 RNAi was expressed in the α'β', α/β and γ lobes, respectively. Bar graphs represent the mean and error bars represent the SEM. Asterisks denote significant difference between groups (* $P < 0.05$, ** $P < 0.01$, *** $P < 0.001$). Numbers signify P-values (Welch two sample t-test). (B) Additional RNAi lines targeting unc-5 were used to verify the memory phenotypes. Bar graphs represent the mean and error bars represent the SEM. Asterisks denote significant difference between groups (* $P < 0.05$, ** $P < 0.01$, *** $P < 0.001$). Numbers signify P-values (Welch two sample t-test).

## Discussion

The consolidation into more stable long-term memories (LTMs) requires the activation of molecular programs for the "de novo" production of proteins [1–4]. Some of these proteins may be involved in synaptic plasticity processes, such as alterations of dendritic arborization and synaptic bouton density, reinforcement of synaptic transmission, trafficking of synaptic vesicles [7,58].

Addressing CrebB transcriptional regulation may constitute an entry point to the mechanistic aspects of LTM. Since CrebB operates in conjunction with different co-factors during LTM establishment and maintenance [19], it is intuitive that also the genetic program may change according to a certain phase. This study focused on the identification of memory-phase specific CrebB target genes with the aim to characterize their potential roles as LTM phase regulators.

### Targeted DamID to profile CrebB targets

To profile cell-type specific DNA-binding of CREB with a spatial and temporal resolution, we made use of the Targeted DamID (TaDa) method [42]. The final selection of candidate CREB targets was done considering a list of genes including CREB/CRTC-regulated CRE sites available in a study published by Hirano et al., 2016 [19]. In the mentioned study, the authors performed ChIP-seq experiments following aversive LTM formation. Aversive and reward LTM share some similarities, such as the induction of CREB-mediated transcriptional regulation and the requirement of the synthesis of new proteins [24,59]. While one cycle of reward training is sufficient to form appetitive LTM, aversive training requires repetitive associations to form LTM. Targeted DamID provides an alternative technical approach to identify the genes, which allows anatomically restricted analysis. An advantage compared to ChIP-seq is that TaDa protein-DNA profiling occurs *in vivo* and does not require protein fixation, which may cause artefacts. While ChIP-seq requires large starting materials, TaDa can be performed with fewer cells. For this study, for example, 50–100 whole heads per sample were used (~200,000 cells per head). TaDa can provide binding coordinates over time intervals, while ChIP-seq profiles a snapshot of the protein-DNA interaction. However, TaDa relies on the frequency of GATC sites, which occurs, on average, every 200 bp. For this reason, it a has lower resolution than ChIP-seq. TaDa is an adaptation of the DamID and, in principle, it allows to profile any TF-DNA interaction. However, it has been optimized and mainly implemented for Pol-II binding and no other studies have ever performed CREB-TaDa profile so far. CREB nuclear localization depends on its activation state. Different kinases phosphorylate serine residues at the level of the CREB KID domain. It is believed that the interaction with co-factors and post-translational modifications, such as phosphorylation, methylation, acetylation, ubiquitination and SUMOylation, influences both the activity and the nuclear-cytoplasmatic distribution of CREB proteins [60,61]. Thus, due to the multitude of regulatory mechanisms, characterizing the temporal and spatial dynamics of CREB remains challenging.

During the training only a few KCs are involved in learning a specific odor response. Thus, binding of Dam::CrebB to its target genes and methylation of nearby GATC sites will only happen in a small subset of the KCs. Therefore, the identification of potential CrebB targets may be impaired by the dilution of methylated sites in the majority of KCs, that do not respond to the specific odor used. In spite of this dilution effect, we were able to identify potential CrebB targets and further verify a role for them in LTM formation. To determine if our candidates are direct targets of CrebB we would have to identify the actual binding sites and demonstrate that binding of CrebB will result in a change in gene expression.

We chose the mb247-Gal4 driver to induce Dam::CrebB expression specifically in the MB cells. Alternative drivers like ok107-Gal4 can be broader, labelling cells that reside outside the MBs and increasing the probability to obtain a toxic effect.

## Candidate CrebB-target genes

To validate the involvement of the candidate CrebB targets in LTM, we set up a screen to test 24h or 48h memory in the respective RNAi lines. This screen allowed us to select *HERC2*, *cic*, *esn* and *unc-5* as candidate genes affecting LTM.

It is important to consider, however, the potential limitations connected to an RNAi screen. To give some examples, the efficiency of the knock-down could vary between different RNAi constructs; some RNAi constructs affect off target genes or have leaky expression which could potentially interfere with development or other biological functions. HCR detection of the mRNA was performed to validate the expression of these genes in the MBs. While *HERC2*, *cic* and *unc-5* mRNA were detected in MB cells, *esn* is only expressed in a few cells. These analyses were consistent to single cell transcriptomic data [50]. Even though *esn* does not show constitutive expression in the MB cells, its hypothetical transient expression could be the effect of learning. However, one concern regards the quality of the RNAi line tested. The RNAi construct used in this study to knockdown *esn* (VDRC 32040) has two potential off-targets (CG32053, a predicted hexose transmembrane transporter, and CG17636, a glutathione gamma-glutamate hydrolase) which could be the actual link to the observed memory defects. All four candidate genes have been proposed to play a role in synaptic plasticity.

## HERC2

HERC2 has a predicted "ubiquitin-protein ligase" function and, as an E3 member of the ubiquitin proteasome system, is involved in protein degradation through the proteasome-mediated pathway. Ubiquitin signalling pathways hold important implications for the regulation of neuronal connectivity and plasticity in the brain. Impairments at the level of this system have been associated to neurodegenerative and neurodevelopmental decline, such as Alzheimer disease and autism [62–64]. While some E3 ubiquitin ligases, such as Cdc20-AP, promote synaptic proliferation [65], others suppress synaptic boutons in the brain [62,66]. A recent study has revealed HERC2 involvement in the regulation of synaptic formation [66]. HERC2 interacts with the autism-linked ubiquitin ligase RNF8/UBC13 and the scaffold protein NEURL4, taking part in a novel cytoplasmic ubiquitin-signalling network that suppresses synapse formation in the brain. RNF8/UBC13 knockout showed impaired cerebellar-dependent learning in mice. Similarly, in *Drosophila*, ubiquitin ligase UBE3A plays a role in synaptic suppression and its mutation showed increased number of presynaptic boutons at neuromuscular junctions [62]. These data do not seem in line with our behavioural phenotype, as HERC2-RNAi showed an increased 24h LTM performance. However, in a large RNAi screen where 3200 RNAi lines were tested for 3h memory, HERC2 knockdown showed an enhancement in memory score [28] and our results support its involvement as a negative regulator of memory.

## cic

*cic* (*Capicua*) encodes one of the transcriptional repressors of the high mobility groupbox (HMG-box) family. It is regulated by the receptor tyrosine kinase (RTK) signalling pathway, downstream of the Ets transcription factor family [67]. Previous studies in *Drosophila* have described *cic* for its role during the differentiation of the embryo termini and

organ growth [68]. During brain development in mice, *cic* seems to contrast normal neuronal differentiation as it impairs the transition of neuroblasts to immature neurons in the hippocampus [69]. Human research also has investigated *cic*-mediated regulation as its mutation is commonly detected in human cancer and neurodegenerative diseases [70–72]. Beside its involvement in development, *cic* is highly expressed in differentiated, adult neurons. Loss- and gain-of-function studies have revealed that Cic suppresses dendrite formation and growth inhibiting the transcription of *Ets*, in mice hippocampal neurons [73]. Precise dendrite distribution and synaptic coordination are critical for a proper functional neural activity. Interestingly, learning and memory ability in mouse was impaired when disrupting cic-Ataxin1 interaction with consequent misallocation of neocortex neurons [74]. Our results are consistent with previous behavioural observations in mice, supporting *cic* involvement in memory. The knockdown of *cic* has resulted in 0h and 24h memory defects.

## unc-5

Netrins are chemotropic cues which guide cell migration and axon extension during development. Following development, netrins and their receptors continue to be expressed in neurons and recent studies have discovered their implication in synaptic plasticity, learning and memory [57]. In mice, Netrin-1 and its receptor DDL are enriched at synapses of hippocampal neurons and specific knockout impairs spatial memory [56]. Another evidence in mice showed netrin-1 potential to recover amyloid-β induced memory impairment, during late-phase of Long-Term Potentiation, a process by which synaptic connections between neurons become stronger with frequent activation [55]. The binding of Netrin-1 to DCC seems to be required for plasticity at hippocampal synapses, via the activation of a signalling cascade downstream of NMDAR [75]. In humans, genetic polymorphism in netrin-1 and its receptors have been linked to neurodevelopmental and neurodegenerative disorders. In *Drosophila*, netrins, and other axonal guidance molecules, regulate the morphogenesis of the MB lobes. Structural integrity of the MB is essential for learning and memory. The main receptors of netrins in flies are Unc-5, Fra (Frazzled) and Dscam (Down syndrome cell adhesion molecule). After knockdown of *unc-5*, the dorsal or medial lobes were partially or completely lost compared with control [54]. Morphological defects of the MB have been linked to impaired memory and sleep deficit. In our screen *unc-5* RNAi impaired 48h LTM. 0h and 24h memory tests supported a role for Unc-5 also during memory formation and early maintenance. Restricting Unc-5 knockdown within MBs lobes did not dissect unc-5 activity any further as all the 3 MB-lobes drivers used lead to impaired 24h LTM. To exclude any potential developmental effects, we restricted *unc-5* knockdown to the adult stage by using tub-Gal80ts. The results presented in Fig 4 indicated that unc-5 is indeed involved in LTM-regulatory mechanisms, apart from its developmental implication.

Despite having a clear behavioral phenotype, the qPCR does not indicate a significant reduction in *unc-5* levels. One explanation would be that for the behavioral assay *unc-5* was reduced only in the mushroom body affecting specifically the cells required for LTM formation whereas for the qPCR expression of the RNAi was driven with the pan-neuronal elav-Gal4 driver and whole heads were used for RNA extraction. Therefore, non-neuronal tissue might mask or compensate the loss of unc-5 in neurons. For instance, *unc-5* is also expressed in glia cells [76] and is involved in short-range repulsion during motor axon guidance [77]. Nonetheless, the fact that we observe the same memory phenotype with three different *unc-5* RNAi-lines targeting non-overlapping regions of the *unc-5* mRNA, strongly supports a specific requirement of Unc-5 in LTM stability.

### esn

*esn* (espinas) encodes for a LIM domain protein which binds to stan, a transmembrane cadherin. The stan-esn interplay regulates the repulsion between dendritic branches of sensory neurons. LIM homeodomain proteins are transcription factors and exert their effects by regulating target gene expression. While little is known about esn, other LIM homeaodomain proteins, Apterous and its cofactor Chip, seems to be required for LTM maintenance [78].

If we assume, that all four candidates are regulated by CrebB, *cic* and *unc-5* expression would be induced by CrebB activation, which would increase LTM formation. Thus, knockdown of these 2 genes should result in loss of LTM, which is what we see. However, cic and unc-5 knockdowns show already a learning defect at 0h, which is in accordance with the fact, that they are constitutively expressed in MB cells. The reduced LTM may therefore be a result of reduced learning efficiency that is likely to occur independent of CrebB. HERC2 seems to be required for forgetting since knockdown of HERC2 results in increased LTM. If HERC2 gets directly activated by CrebB, knocking it down would improve LTM formation, which is what we see in the RNAi test. For the same reason esn would have to be downregulated to allow LTM formation. A putatively LTM-specific function may be assigned to *esn*. Since *esn* appears to not to be expressed in MBs based on HCR and single-cell RNAseq data, there is no point silencing it and thus one would not expect to get a learning defect that is monitored directly after training. However, if CrebB activation leads to a transient induction of esn in MBs, its upregulation might repress LTM formation, while knocking it down in this condition (after it has been induced in a training and CrebB dependent fashion) would enhance LTM.

This study has revealed learning and memory genes, based on their differential expression in two memory intervals and their learning phenotype in MB knockdown experiments. Follow up investigations, however, will be required to resolve some of the limitations and provide insight into the mechanism of how the identified genes. Restricting the expression of RNAi to adult stage, for instance, would exclude the potential implication of developmental defects of the MB.

## Methods

### Fly husbandry

*Drosophila melanogaster* flies were reared in plastic vials on standard cornmeal food supplemented with fructose, molasses, and yeast, and transferred to fresh food vials every 2 to 3 days. Flies were generally kept at 25°C, 65% humidity, and exposed to 12 h light– 12 h darkness cycle. Crosses with tubGal80$^{ts}$ were raised at 18°C and moved to 29°C, according to the experimental procedure.

MB-driver mb247-Gal4 was obtained from Dennis Pauls (University of Würzburg). UAS-RNAi lines used for the screen were received from the VDRC stock center [49] or Bloomington Drosophila Stock Centre (NIH P40OD018537) (see S1 Table for stock numbers). Based on the background of the RNAi lines, w$^{1118}$, y$^1$w$^1$, y$^1$v$^1$ or y$^1$sc*v$^1$sev$^{21}$ lines were chosen as parental controls. 5-HTR1B-Gal4 (27636), UAS-Dcr-2 (24648), and tub-Gal80$^{ts}$ (7019) were obtained from the Bloomington stock center. c739-Gal4 from R. Tanimoto. VT030604-Gal4 (200228) from VDRC. The experimental lines were constituted by the offspring of the crosses between UAS and Gal4 lines. As control, the parental Gal4-lines were crossed with w$^{1118}$, y$^1$w$^1$, y$^1$v$^1$ or y$^1$sc*v$^1$sev$^{21}$ lines, according to the genomic background of the respective RNAi lines.

## Generation of UAS-Dam::CrebB expressing flies

The CrebB cDNA clone (RT01110, Flybase Id FBcl0383842) was obtained from the Drosophila Genomic Resource Center (DGRC Stock number 1623195). It contains the alternatively spliced exons present in about half of the CrebB isoforms. The coding sequence was amplified using primers "CrebB Start NotI fw" (GAGCGGCCGCACATGGACAACAGCATCGTC-GAG, NotI restriction site underlined) and "CrebB Stop XbaI re" (GCTCTAGAAGCTTT-CAATCGTTCTTGGTCTGACAG, XbaI restriction site underlined). The PCR-fragment was cloned NotI-XbaI into the "pUASTattB- LT3-NDam" vector [42] in frame with the Dam coding sequence. The plasmid was injected into $y^1w^1$, nos-PhiC31; attP40 (Bloomington stock 79604) eggs for site directed integration on the $2^{nd}$ chromosome.

## Learning machine

For behavior experiments, we used a memory apparatus that is based on Tully and Quinn's design and modified it to allow conducting 4 memory experiments in parallel (CON-Elektronik, Greussenheim, Germany). Experiments were performed at 25˚C and 65% to 75% relative humidity. The training was performed in dim red light. The 2 odors used were 3-Oct (Sigma-Aldrich Cat# 218405-250G; CAS Number: 589-98-0) and MCH (Sigma-Aldrich Cat# 66360-250G; CAS Number 589-91-3) diluted in paraffin oil (Sigma-Aldrich Cat# 18512–2.5L; CAS Number 8012-95-1) to 3:100 and 8:100, respectively. Final volume of 260 μL of the diluted odors were presented in a plastic cup of 14 mm in diameter. A vacuum membrane pump ensured odor delivery at a flow rate of 7 l/min.

## Appetitive olfactory conditioning

Before appetitive conditioning, groups of 50 to 100 flies with mixed sexes were starved for 18 to 21 h in plastic vials containing cotton damped with distilled water at the bottom. Experiments were excluded when more than half of the flies were not healthy/dead, there were technical problems with the machine or human errors happened. Position in the machine and the sequence in which the genotypes were tested were randomized. The training was conducted at midday. The conditioning protocol consists of a 90 s accommodation period, 120 s of the first odor, 60 s of fresh air followed by 120 s of the second odor. During the first odor, flies were in a conditioning tube lined with filter paper that was soaked in distilled water the day before the experiment and left to dry overnight at RT. For the second odor, flies were transferred to a conditioning tube lined with a filter paper that was soaked with a 1.5 M sucrose (Sigma-Aldrich, Cat# 84100-1KG; CAS Number 57-50-1) solution on the day before and left to dry overnight at RT. After conditioning, flies were either directly tested for STM or put back in starvation vials until the memory test 24 h or 48 h later. For 48 h memory, flies were fed for 3/4 h after training, before starving them again. Each experiment consisted of 2 conditioning sessions, in which the odor paired with sucrose was reversed.

## Memory test

Flies were loaded into a sliding compartment and transferred to a two-arm choice point. Animals were allowed to choose between 3-Oct and MCH. After 120 s, flies trapped in both arms were collected separately and counted. Based on these numbers, a preference index was calculated as follows:

$$\text{Preference Index PI} = ((N_{arm1} - N_{arm2})\,100)/N_{total}$$

The 2 preference indices were calculated from the 2 reciprocal experiments. The average of these 2 PIs gives a Learning Index (LI).

$$\text{Learning Index (LI)} = (PI1 + PI2)/2.$$

## Sensory test

Flies were tested for their ability to sense the 2 odors 3-Oct and MCH and their preference/ appetite toward sucrose. Therefore, the flies were loaded into a sliding compartment and brought to a two-arm choice point. The flies were allowed to freely choose between an arm containing the stimulus and a neutral arm. All experiments were carried out in the dark. Afterward, the flies in each arm were counted, and a preference index was calculated.

For testing the odor response, the flies could choose between one of the odors in the same concentration as used for the behavior experiment and the same amount of paraffin oil for 120 s. This test is referred to as "odor avoidance".

$$\text{Preference Index PI} = ((N_{air} - N_{odor})\,100)/N_{total}.$$

For testing the odor preference, the flies could freely choose between the two odors, presented concomitantly and at the same concentration as used for behavior experiment for 120 s. This test is known as "odor preference".

$$\text{Preference Index PI} = ((N_{odor1} - N_{odor2})\,100)/N_{total}$$

For testing sugar sensitivity, a group of flies was starved for 19 to 21 h in a tube with damp cotton on the bottom. They could choose for 120 s between a tube lined with filter paper that was soaked in 1.5 M sucrose solution the day before or a tube lined with filter paper that was soaked in distilled water the day before. This is the "sucrose response" test.

$$\text{Preference Index PI} = ((N_{sucrose} - N_{water})\,100)/N_{total}.$$

## Statistics

To compare performance indices, we used one-way analysis of variance (ANOVA) with post hoc Tukey honestly significant difference (HSD) test calculator for comparing multiple treatments in R with the package multcomp. In the case of 2 groups, we performed a t test for comparison.

## Targeted DamID

Samples were prepared similarly to previous TaDa experiments described in Widmer et al., 2018 [29]. UAS-Dam and UAS-Dam-CrebB lines were crossed to tubGal80ts; mb247-Gal4. The offspring was reared for 7/8 days at 18˚C and trained with the paired or unpaired olfactory appetitive conditioning paradigm. Expression of Dam and Dam-CrebB was induced by shifting the flies to 29˚C in two main time intervals (TIs). For TI-1, flies were moved to 29˚C 3h before conditioning, trained at 25˚C, and moved back to 29˚C for 24 hr. For TI-2, animals were trained at 18˚ and shifted to 29˚C 24–48h after conditioning. At the end of the 29˚C-time interval, flies were frozen in liquid nitrogen and heads were collected. Extraction of genomic DNA from the fly heads (50–100 per sample), amplification of methylated fragments, DNA purification, and sonication was performed according to Marshall et al. 2016 [43]. After

sonication, DamID adaptors were removed by digesting overnight at 37˚C with 5 units of Sau3AI (R0169S; NEB). The sequencing libraries were prepared according to the Illumina Tru-Seq nano DNA library protocol. The samples were sequenced using NGS (Illumina HiSeq3000) at an average of ∼15 million paired-end reads per sample.

## Targeted DamID analysis

Follow up analysis of the FASTQ files, obtained from the sequencing experiment, were done in a paired-end subset (R1), processed with the damid pipeline, as previously described ([48]), and mapped to release 6.22 of the Drosophila genome [79]. The damid pipeline performed sequence alignment, read extension, binned counts, normalization and pseudocounts. To determine the list of significantly expressed genes for each condition, we used a cut-off of 0.2 for the log2 fold-change (Dam-CrebB over Dam-only) [42]. The adjusted p-values (q-values), were assigned based on the signal from multiple GATC fragments and using a 0.01 false discovery rate (FDR) threshold. Following the extraction of the significant genes among the three replicates for each condition (paired, unpaired, and control), the list of paired genes was compared with the unpaired one using the comparison pipeline. CREB-targets from a previous ChIP-seq study [19] were taken into account for the selection of the final candidate genes.

## In situ hybridization HCR

*Drosophila* brains were dissected in phosphate buffer (PBS) and fixed in 4% formaldehyde for 20 min at room temperature. Samples were prepared for hybridization chain reaction (HCR) RNA fluorescent in situ hybridization (RNA-FISH) following the Ferreira et al. 2020 protocol [80], available in protocols.io (https://www.protocols.io/view/hcr-rna-fish-protocol-for-the-whole-mount-brains-o-81wgb7joyvpk/v1). Probes were designed and purchased from Molecular instruments, together with HCR reagents. The HCR protocol used was adapted from the provided "generic sample in solution" protocol (https://files.molecularinstruments.com/MI-Protocol-RNAFISH-GenericSolution-Rev8.pdf, Choi et al., 2018 [81]) with the following modifications: samples were washed three times in PBST for 15min at room temperature, before prehybridization. Probe solution was prepared at a concentration of 16nM by adding 4 pmol of each probe mixture to warm Probe Hybridization Buffer with a final volume of 250μL. Incubation time for hybridization was increased to 24h to enhance signal. Following washes with Probe Wash buffer (5x 10min), samples were washed 2x5min with 5X-SSCT before pre-amplification. Incubation time for amplification was increased to 24h to enhance the signal. Finally, to remove the excess of hairpins, samples were washed 1x5min in 5X-SSCT, 1x15min in Probe Wash Buffer, 1x10min is 5X-SSCT and rinsed in 1X Nuclease-Free PBSPBS before overnight incubation in Vectashield antifade mounting medium. The experiments were performed by combining B1- and B2-Alexa-647 hairpins in *Drosophila* brains dissected from the reporter line UAS-myr::GFP/+;mb247-Gal4/+.

## Electrophoretic mobility shift assay

The coding sequences of Dam::CrebB and CrebB alone were cloned into the pBluescript vector. Both plasmids were linearized and used as template for protein production with the TNT (R) T7 Quick Coupled T/T System kit (Promega). The translated proteins were mixed with IRDye 700 CREB consensus oligonucleotide (LI-COR) with or without 10-fold excess of consensus or mutated competitor oligonucleotides and incubated for 30 minutes. The reactions were run on a 5% acrylamide gel (biorad) and the oligonucleotides visualized with an Odyssey FM imaging system (LI-COR).

## qPCR

UAS-RNAi lines for the different candidates were crossed to elav-Gal4 flies to knock-down the expression of the candidate genes in the whole brain. Heads of 30 to 50 flies from each cross were homogenized in Qiazol Lysis reagent (QIAGEN) and the extracted total RNA was used as template for reverse transcription (GoScript reverse transcription system, Promega). For the genes containing introns random primers were used for reverse transcription. For genes that do not contain introns (Pfrx, Sry-delta and slp1) we used oligo d(T) primers to reduce the probability of amplifying genomic DNA during qPCR. The resulting cDNAs were mixed with KAPA SYBR FAST qPCR master mix (Kapa Biosystems) and specific primers (S2 Table). For genes where the RNAi-line was directed against a subset of isoforms, we tested for RNA-levels with both, a primer pair only specific for the isoforms that are affected by RNAi, and with primers that should recognize all isoforms. To compare relative expression levels between different crosses, we used primers for actin42C RNA as standards. qPCR reactions were amplified on a Rotor-gene Q real-time PCR cycler (QIAGEN) and analyzed with Microsoft excel.

## Supporting information

**S1 Fig. Electrophoretic Mobility Shift Assay.** Electrophoretic Mobility Shifts for CrebB and Dam-CrebB binding to CREB-site. Labeled probe is bound by CrebB protein alone and by Dam::CrebB fusion protein resulting in a shift of the signal (shifted probe). Addition of unlabeled oligonucleotides with the same sequence as the probe (consensus competitor) reduces the amount of shifted probe, while addition of unlabeled oligonucleotides with a mutated binding site (mutant competitor) does not reduce the amount of shifted probe. Below the gel the sequences of the labeled probe and consensus competitor oligonucleotide (CREB consensus) and the mutant competitor oligonucleotide (CREB mutant) are shown. Mutated residues in small letters. Binding site underlined.
(TIF)

**S2 Fig. Dam::CrebB and Dam-only 24h and 48h LTM.** UAS-Dam-only or UAS-Dam::CrebB expression was induced in the MB of adult flies using the driver mb247-Gal4 and co-expressing tub-Gal80ts. Transgenic crosses set up at the permissive temperature (18˚C) and adult flies were transferred to 29˚C for3/4 days. Adult flies expressing Dam-only did not show changes in the 24h or 48h memory performance (A-B). Similarly, adult flies expressing Dam::CrebB did not show differential learning scores at the same time points (C-D). Asterisks indicate the P-value of the comparison with control crosses (mb247-Gal4>w1118/ y1w1/y1v1/y1sc*v1sev21) (* $P < 0.05$, ** $P < 0.01$, *** $P < 0.001$); n = 7–10 for MB > RNAi, n = 7–28 for control crosses. Bar graphs represent the mean and error bars represent the SEM.
(TIF)

**S3 Fig. qPCR.** qPCR on RNAi-lines crossed to elav-Gal4 (orange bars) to analyze RNAi-levels in comparison to controls (elav-Gal4 x background, dark grey; and UAS-RNAi lines x background, light grey; expression levels relative to endogenous actin42C RNA levels, arbitrary units).
(TIF)

**S4 Fig. HCR expression in the MB reporter lines.** (A, B, C, D) magnified images of MB cells, showing the mRNA expression of the 4 hits. (A', A") alternative sections of HERC2 mRNA expression (red). (B',B") alternative sections of cic mRNA expression (red). (C',C") alternative sections of esn mRNA expression (red). (D',D") alternative sections of unc-5 mRNA

expression (red). mb247>GFP (green), Dapi (blue).
(TIF)

**S1 File. Data referring to Fig 1D.**
(XLSX)

**S2 File. Data referring to Fig 2.**
(XLSX)

**S3 File. Data referring to Fig 3.**
(XLSX)

**S4 File. Data referring to Fig 4.**
(XLSX)

**S5 File. Data referring to Fig 5.**
(XLSX)

**S6 File. Data referring to S2 Fig.**
(XLSX)

**S1 Table. RNAi lines used for the LTM screen.**
(XLSX)

**S2 Table. qPCR primers.**
(XLSX)

## Acknowledgments

We thank the Next Generation Sequencing Platform of the University of Bern for performing library preparation and sequencing experiments. We would like to thank the Vienna and Bloomington stock centers for fly strains. We would also like to thank colleagues of the Sprecher lab for valuable input and discussions.

## Author Contributions

**Conceptualization:** Noemi Sgammeglia, Simon G. Sprecher.

**Formal analysis:** Noemi Sgammeglia, Yves F. Widmer, Jenifer C. Kaldun.

**Funding acquisition:** Rémy Bruggmann, Simon G. Sprecher.

**Investigation:** Noemi Sgammeglia, Yves F. Widmer, Jenifer C. Kaldun.

**Methodology:** Noemi Sgammeglia, Yves F. Widmer, Jenifer C. Kaldun, Cornelia Fritsch, Rémy Bruggmann.

**Writing – original draft:** Noemi Sgammeglia, Jenifer C. Kaldun.

**Writing – review & editing:** Noemi Sgammeglia, Jenifer C. Kaldun, Cornelia Fritsch, Simon G. Sprecher.

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
