## [Decision Letter · Decision Letter 0]

13 Dec 2022

Dear Dr Sprecher,

Thank you very much for submitting your Research Article entitled 'Memory phase-specific genes in the Mushroom Bodies identified using CrebB-target DamID' to PLOS Genetics.

The manuscript was fully evaluated at the editorial level and by independent peer reviewers. The reviewers appreciated the attention to an important problem, but raised some substantial concerns about the current manuscript. Based on the reviews, we will not be able to accept this version of the manuscript, but we would be willing to review a much-revised version. We cannot, of course, promise publication at that time.

If you decide to revise the manuscript for further consideration at PLOS Genetics, please aim to resubmit within the next 60 days, unless it will take extra time to address the concerns of the reviewers, in which case we would appreciate an expected resubmission date by email to plosgenetics@plos.org.

We are sorry that we cannot be more positive about your manuscript at this stage. Please do not hesitate to contact us if you have any concerns or questions.

Yours sincerely,

Liliane Schoofs

Academic Editor

PLOS Genetics

Gregory P. Copenhaver

Editor-in-Chief

PLOS Genetics

Reviewer's Responses to Questions

**Comments to the Authors:**

Reviewer #1: In this work, Sgammeglia et al generate UAS-Dam::CrebB tool for identifying CrebB target gene expressions in olfactory memory center, the mushroom body, after appetitive conditioning. Combined with tub-GAL80ts transgene allows the authors to distinguish the CrebB target gene expressions within different time intervals (TIs) via shifting the flies from permissive to restrictive temperature during memory formation. Two different TIs protocols were used in their assays, which are 3-hour before training to 24-hour after training(TI-1) and 24 to 48 hour after training (TI-2). The authors also compared with previous ChIP-seq dataset and identified 16 and 17 genes are differentially regulated in TI-1 and TI-2, respectively. RNA interference (RNAi)-mediated gene silencing and behavioral screen identified 4 genes as potential targets of CrebB. Genetic silencing of these 4 genes in mushroom body also disrupted appetitive memories. Overall, this study provide an elegant tool for investigating potential genes under CrebB regulations in specific neurons. However, there are several issues are needed to be resolved before publication. Listed below are my specific comments:

Major comments:

1. The major contribution of this study is based on the Dam::CrebB fusion protein, which allows for identifying the potential CrebB target gene expression in specific neurons (this study focused on mushroom body). However, the effects of genetic modification of CrebB was not checked before their experiments. Whether the Dam::CrebB fusion protein still can bind to CRE as regular CrebB does? The authors should designed experiments to show that CRE binding ability is not affected in the Dam::CrebB as compared to regular CrebB.

2. The experimental design for identifying the candidate CrebB downstream gene expression after paired training includes 2 time intervals (TI-1 and TI-2 in Figure 1B), which shift the temperature from 18 to 29 °C at different periods for silencing GAL80ts activity. However, the permissive temperature condition should be included in this experiments, which is the same flies all the way stay in 18 °C.

3. A lot of UAS-geneRNAi lines target different genes were used in this study, but there is no any verification for the knockdown efficiency of the RNAi lines used in this study.

4. In sufficient controls in several figures. All the behavioral data of the manuscript only use GAL4/+ as the genetic control groups but the UAS-geneRNAi/+ controls are missing in whole manuscript. The UAS-geneRNAi/+ control groups should also be conducted and included in the behavioral data.

5. Since unc-5 is important for the development neurites, therefore RNAi-mediated silencing of unc-5 in mushroom body affects memory may cause by the developmental defects of mushroom body. The authors should combine gene switch tools in their assay (i.e. tub-GAL80ts or MB-GeneSwitch) for bypassing the developmental effects.

6. In Figure 3D, knockdown of esn in mushroom body disrupted The 24- and 48- hour appetitive memories but the esn mRNA is not detectable in mushroom body neurons. The authors should use other techniques (i.e. immunohistochemistry) instead of in situ hybridization to detect the expression of esn in mushroom body.

Minor comments:

1. line 62-62, the it should be: αβ and α'β' KCs projections form vertical and horizontal lobes, whereas γ KCs only form horizontal lobe of MB.

Reviewer #2: Long-term appetitive memory requires CREB-dependent de novo gene expression and protein synthesis. Sgammeglia et al. developed the TaDa technique to identify CerebB target candidate genes during two-time intervals; TI-1 (0~24 hrs after training) and TI-2 (24 ~ 48 hrs after training). The authors assigned TI-1 for memory formation and consolidation and TI-2 for maintenance and found 111 genes in TI-1 and 26 genes in TI-2. Among candidate genes, the authors found that knocking down HERC2 and cic (TI-1) and esn and unc-5 (TI-2) impair memory.

While the authors developed a new gene screening method, due to practical concerns, they do not show how these genes are involved in long-term memory. For example, while they claimed that cic is the TI-1 gene and unc-5 is the TI-2 genes, knockdown of these genes resulted in defects in 0-, 24-, and 48 hr-memories. Therefore, it is unclear whether the functions of these genes are required during the assigned time interval, and whether their expressions are changed.

Major concerns

1. Temporal gene knockdown. It is unclear whether observed memory defects are caused by acute dysfunction of candidate genes or developmental defects. Authors should knock down candidate genes at least at the adult stage using temporal knockdown methods, such as TARGET and Gene Switch system. As mentioned above, 0 hr memory impairment in cic and unc-5 mutants could be due to defects in neural development. Furthermore, while authors exclude RNAi lines that have problems with the viability and health of MB>RNAi offspring, temporal knockdown may resolve these issues and increase the number of genes related to LTM.

2. Genomic background. UAS-RNAi and Gal4 driver lines should be outcrossed with their UAS-RNAi parental control lines. The authors compared the memory of knockdown lines with their parental background lines. However, the reviewer thinks it is not sufficient. Every line in the lab stock has spontaneous mutations over time. Therefore, the genetic background of the parental line in the stock center is likely to be different from the author's lab.

3. Absence of quantification of transcripts. The authors did not attempt to examine whether TI-1 and TI-2 genes actually changed their expression during these time intervals upon LTM formation.

Minor comments

Functional classification and annotation of 111 TI-1 and 26 TI-2 genes would be more informative for people in the research field.

Reviewer #3: This manuscript uses a CREB-DAM fusion followed by genetic screening to identify novel regulators of long-term memory. Long term olfactory memory is model that has received enormous attention, with the foundational studies showing fundamental differences between short and long-term memory taking place nearly 30 years ago. While the role of CREB is LTM is now known to be deeply evolutionarily conserved, the role of factors downstream of CREB is still incomplete. The olfactory training experiments are well controlled, allowing for comparison of CREB-DAM results from trained and untrained flies. Overall, the approach is novel and identifies multiple new genes that are involved in memory. The experiments are well controlled and it is a significant advance for the field. Additional mechanistic follow-up would increase the impact of this manuscript, though I do not believe it is essential for the initial description of the screen. Overall this is a strong manuscript on a topic that will be of broad interest to the community and the new genes identified provide a foundation for future studies.

1. The experimental protocol overexpresses CREBB. It’s an important control to show that in the contexts of these manipulations (including temperature shifts) there is no effect on memory.

2. A general point worthy of more discussion is the choice of MB-driver. In addition, these memories may be encoded sparsely in Kenyon cells, which may dilute the impact of the technique. This does not invalidate the results in any way, but is worthy of discussion.

3. Line 164: It is surprising that health RNAi knockdown flies is an issue since this brain region is non-essential. This is likely due to non-specific expression of the driver. Is it possible to test with another driver?

4. The HCR data in Figure 3 is insufficient. Perhaps magnified images of individual cells would more clearly highlight the localization? One drawback is only a fraction of the MB cells are shown. Is it possible to predict whether the identified genes are expressed in all MB neurons or only a fraction?

5. Why was the choice made to focus on unc5 of all the hits?

6. Given all the data on Kenyon cell subtype specificity in memory it is surprising that unc5 knockdown has an effect in all cell types. Why might this be?

Line 48: Memory consolidation also refers to protein-synthesis independent memories that stabilize following learning. To clarify, I would explicitly state ‘protein synthesis-dependent long-term memories.

Line 68-72: I presume these differences in CREB are simply because it serves the same functions in circuits dedicated to different types of memory.

Line 124: Suggest not referring to the control group as Naïve, because the have been trained…simply in a way that does not confer memory.

The lines showing significance between controls and experimental are unclear. Simply showing the star would be clearer.

Line 271: Is esn expression very low, or not expressed?

Line 349: The statement is a big assumption.

**Have all data underlying the figures and results presented in the manuscript been provided?**

Reviewer #1: None

Reviewer #2: Yes

Reviewer #3: Yes

PLOS authors have the option to publish the peer review history of their article (what does this mean?). If published, this will include your full peer review and any attached files.

Reviewer #1: No

Reviewer #2: No

Reviewer #3: No

---

## [Decision Letter · Decision Letter 1]

28 Apr 2023

Dear Dr Sprecher,

Thank you very much for submitting your Research Article entitled 'Memory phase-specific genes in the Mushroom Bodies identified using CrebB-target DamID' to PLOS Genetics.

The manuscript was fully evaluated at the editorial level and by independent peer reviewers. The reviewers appreciated the attention to an important topic but identified some concerns that we ask you address in a revised manuscript.

We therefore ask you to modify the manuscript according to the review recommendations. Your revisions should address the specific points made by each reviewer.

Yours sincerely,

Liliane Schoofs

Academic Editor

PLOS Genetics

Gregory P. Copenhaver

Editor-in-Chief

PLOS Genetics

Reviewer's Responses to Questions

**Comments to the Authors:**

Reviewer #1: The authors have addressed most of my previous comments. However, I still have one concern regarding the RNAi knockdown efficiency of unc-5 is not very significant in Q-PCR data (see Figure S3), but significant behavioral phenotype was shown when unc-5-RNAi expression in mushroom body (see Figure 2G, 3C, 4D, 5). The author should at least have some discussions/explanations regarding the possibility of these inconsistent results in the manuscript.

Reviewer #3: The authors ahve addressed all my concerns in the revised version. There are a number of places where critical data was added including additional controls for the Dam-CrebB construct and additional HCR analysis.

**Have all data underlying the figures and results presented in the manuscript been provided?**

Reviewer #1: Yes

Reviewer #3: Yes

PLOS authors have the option to publish the peer review history of their article (what does this mean?). If published, this will include your full peer review and any attached files.

Reviewer #1: No

Reviewer #3: No

---

## [Editor Report · Decision Letter 2]

29 May 2023

Dear Dr Sprecher,

We are pleased to inform you that your manuscript entitled "Memory phase-specific genes in the Mushroom Bodies identified using CrebB-target DamID" has been editorially accepted for publication in PLOS Genetics. Congratulations!

Yours sincerely,

Liliane Schoofs

Academic Editor

PLOS Genetics

Gregory P. Copenhaver

Editor-in-Chief

PLOS Genetics

Comments from the reviewers (if applicable):

**Data Deposition**

http://datadryad.org/submit?journalID=pgenetics&manu=PGENETICS-D-22-01231R2

**Press Queries**

---

## [Editor Report · Acceptance letter]

7 Jun 2023

PGENETICS-D-22-01231R2 

Memory phase-specific genes in the Mushroom Bodies identified using CrebB-target DamID 

Dear Dr Sprecher, 

We are pleased to inform you that your manuscript entitled "Memory phase-specific genes in the Mushroom Bodies identified using CrebB-target DamID" has been formally accepted for publication in PLOS Genetics! Your manuscript is now with our production department and you will be notified of the publication date in due course.

With kind regards,

Zsofia Freund

PLOS Genetics

On behalf of:
